# Analytical Method Optimization of Tetrodotoxin and Its Contamination in Gastropods

**DOI:** 10.3390/foods12163103

**Published:** 2023-08-18

**Authors:** Jian-Long Han, Lei Zhang, Ping-Ping Zhou, Jiao-Jiao Xu, Xiao-Dong Pan, Pei Cao, Xiao-Min Xu

**Affiliations:** 1Zhejiang Provincial Center for Disease Control and Prevention, Hangzhou 310051, China; jlhan@cdc.zj.cn (J.-L.H.); jjxu@cdc.zj.cn (J.-J.X.); xdpan@cdc.zj.cn (X.-D.P.); 2China National Center for Food Safety Risk Assessment, Beijing 100026, China; zhanglei@cfsa.net.cn (L.Z.); zhoupingping@cfsa.net.cn (P.-P.Z.)

**Keywords:** tetrodotoxin, gastropods, *Neverita didyma*, cation exchange clean-up, liquid chromatography-tandem mass spectrometry

## Abstract

Tetrodotoxin (TTX) is an extremely potent marine biotoxin. An analytical method was developed for both trace contamination and extremely high levels of TTX in gastropods by liquid chromatography-tandem mass spectrometry (LC-MS/MS) with clean-up of cation exchange solid phase extraction (SPE) in this study. The limit of detection (LOD) in the sample matrix was 0.5 μg/kg. With the calibration of a screened internal standard (validamycin, IS), the linear range was 0.1–100 ng/mL (1.5–1500 μg/kg in sample matrix) with a correlation coefficient of *r*^2^ > 0.999. The average recoveries at three spiking levels (1.5 μg/kg, 44 μg/kg, and 1500 μg/kg) were 82.6–94.4% with relative standard deviations (RSDs) less than 8.4%. TTX levels in seven gastropods (741 samples) were studied. The contamination and analogues in *Neverita didyma* (*N. didyma*, 565 samples collected in Zhejiang province, China, from 2016 to 2022) were first reported. The detection rate of TTX in *N. didyma* was 34.2%. The average concentration was 23.1 μg/kg, and the maximum value was 2327 μg/kg. The time distribution study indicated that high contaminations of TTX occurred from May to August for *N. didyma*.

## 1. Introduction

Tetrodotoxin (TTX) is an extremely potent marine biotoxin. A global retrospective study by Guardone et al. [1] found that poisoning incidents occurred annually worldwide due to the ingestion of marine animals containing high concentrations of TTX. The marine animals included puffer fishes, gastropods, arthropods, and cephalopods. TTX poisoning caused by gastropods ranked second (20.9%) besides that by puffer fishes (59.9%). The poisonous gastropods included *nassarius* [2], sea slug [3], and trumpet shell [4,5]. Additionally, trace levels of TTX were found in seafoods such as bivalve mollusks and gastropods [6].

The oral median lethal dose (LD_50_) for TTX was 232 μg/kg [7], and the LD_50_ value was 10.7 μg/kg for intraperitoneal administrations in mice [8]. A dose leading to adverse effects was 0.2 mg, and the minimum lethal dose for adults was 2 mg [9]. Japan has set an official regulatory limit of 10 mouse units/g (MU/g) for TTXs in puffer fish tissue (equivalent to 2000 μg/kg) based on mouse bioassay (MBA) [6]. In 2017, based on the intake of shellfish meat with 400 g, the European Food Safety Authority (EFSA) proposed a provisional concentration below 44 μg TTX equivalents/kg shellfish meat, which was considered not to result in adverse effects in humans [9]. The literature focuses more on the study of trace TTX in bivalve mollusks and the toxic levels in *nassarius* [1,6,10]. However, there is limited research on the analytical method and contamination of trace levels of TTX in gastropod seafood. The difficulty in the research of TTX analytical methods in gastropods is the need to consider both trace and highly toxic levels.

MBA is a classic method for TTX at mg/kg level in poisoning samples, while the sensitivity is insufficient for TTX at μg/kg level in seafood [6]. Immunoassay can quickly screen TTX in seafood, but its shortcomings in qualitative confirmation cannot be ignored as well [11]. The instrumental method based on the chromatographic separation principle is the most commonly used method for qualitative and quantitative determination of TTX in seafood [10]. TTX is a water-soluble and polar substance containing multiple hydroxyl groups (Figure 1). The reported method by gas chromatography-mass spectrometry (GC-MS) included two steps of complicated chemical reactions [12] and was not suitable for fast and batch measurement. There are no ultraviolet and fluorescence groups in the structure of TTX (Figure 1). Derivatives suitable for fluorescence detection can be obtained by treatment with 4 mol/L of sodium hydroxide (NaOH) at a high temperature. Combined with the post column derivatization technique, a high performance liquid chromatography-fluorescence detector (HPLC-FLD) was used to measure TTX in puffer fish [13]. Subsequently, this technique was applied for the determination of TTX in *nassarius* samples during poisoning events [2,14]. Liquid chromatography-tandem mass spectrometry (LC-MS/MS) is the preferred method for the determination of TTX in seafood due to its high sensitivity and selectivity [10,15]. However, compared to fish samples dominated by muscle tissue, the gastropod contains a large amount of visceral tissue. Meanwhile, the water content in gastropods is much lower than that in bivalve mollusks. Compared to that in fish and bivalve mollusk samples, the matrix interference is more severe for the determination of TTX by LC-MS/MS in gastropods [15].

Solid phase extraction (SPE) with a C18 cartridge [2] or C18 cartridge combined with ultrafiltration tube separation [14] was used for the clean-up of TTX in gastropods. Only some of the fat-soluble sample matrix and macromolecular proteins can be removed by the C18 cartridge and ultrafiltration tube, respectively. The signal suppression effects of mass spectrometry for TTX cannot be significantly improved by both clean-up methods. Graphitized carbon SPE cartridges are capable of desalting and were used for the clean-up of TTX in bivalve mollusks [10,16,17]. When cleaned by the ENVI-Carb SPE cartridge, the matrix effects of TTX in the adductor muscle of Pacific Oysters was 83%, while the result in the digestive gland was only 55% [18]. Severe matrix suppression effects of TTX were found in visceral tissues. The immune affinity SPE cartridge can selectively adsorb TTX and effectively remove matrix interferences. It is suitable for sample pretreatment of low levels of TTX in seafood [19,20]. However, due to the limitation of column capacity, it is not suitable for the clean-up of high concentrations, especially toxic levels of TTX. Furthermore, it only specifically adsorbs TTX and is not suitable for the determination of TTX analogues. The cation exchange SPE cartridge has been applied for the clean-up of TTX in biological matrices [2,21]. Two factors affecting the TTX recovery should be considered in the food matrix, namely the competitive adsorption of the interfering matrix during the clean-up of cation exchange and the high concentration of hydrochloric acid (HCl) in the eluate [22]. 

We established a new method for the determination of TTX in gastropods by LC-MS/MS after optimizing the sample extraction process, SPE clean-up conditions, and screening of a new internal standard. It was applied to the study of contamination, time distribution, and traceability of TTX in commercially available gastropods.

## 2. Materials and Methods

### 2.1. Samples

*Neverita didyma* (*N. didyma*), *Bullacta exarata* (*B. exarata*), *Rapana bezoar* (*R. bezoar*), *Margarites pupillus* (*M. pupillus*), *Reishia clavigera* (*R. clavigera*), and *Hemifusus tuba* (*H. tuba*) samples were collected from 2016–2022 in the markets of Zhejiang province, China. *Nassarius semiplicatus* (*N. semiplicatus*) samples were gotten from the coast of Zhejiang province, China.

### 2.2. Reagent and Materials 

All reagents and solvents were of analytical grade unless specified. 

TTX was supplied by Anpel (100 mg/L, Shanghai, China); validamycin (internal standard, IS) was bought from Dr. Ehrenstorfer (100 mg/L, Augsburg, Germany); voglibose was supported by TRC Co., Ltd. (Toronto, ON, Canada); the cation exchange cartridge (OASIS MCX, 60 mg/3 mL, treated with 3 mL methanol and water before sample loading) was bought from Waters (Milford, MA, USA). HCl was supplied by Sinopharm Chemical Reagent Co., Ltd. (Shanghai, China); HPLC-grade ammonia (>28%), formic acid, and acetic acid (HAc) were supported by Thermo Fisher Scientific (Shanghai, China); HPLC-grade acetonitrile and methanol were from TEDIA (Fairfield, OH, USA); ultra-pure water was prepared by Millipore (Bedford, MA, USA).

### 2.3. Preparation of Standard Solutions

TTX spiking standard solutions with 10 μg/L and 1000 μg/L were prepared by gradually diluting the stock solution with 50% acetonitrile/water. IS spiking solution with 1000 μg/L was prepared by gradually diluting the stock IS solution with 50% acetonitrile/water. Standard serial solutions with 0.1–100 μg/L of TTX were prepared by diluting the spiking standard solutions with 50% acetonitrile/water. Each milliliter of standard solution was spiked with 10 μL of IS spiking solution.

### 2.4. Instrumentation Conditions

LC-MS/MS with a positive electrospray ionization (ESI^+^) source (8060 NX LC-MS, Shimadzu, Japan) was used in this study. LC separation was performed on an XBridge^TM^ BEH Amide column (3.0 × 150 mm, 1.7 μm, Waters, USA) with a flow rate of 0.4 mL/min. The injection volume was 2.0 μL. The oven was maintained at 30 °C. The mobile phase consisted of 0.1% (*v*/*v*) of formic acid in water (Channel A) and acetonitrile (Channel B). Gradient elution was programmed as follows: 0–1.0 min, 75% B; 1.0–6.0 min, 75–45% B; 6.0–7.0 min, 45% B; 7.0–7.5 min, 45–75% B; 7.5–11.0 min, 75% B. The interface voltage was 4.5 kV. The dissolvent line, MS interface, and heat block temperatures were 250 °C, 300 °C, and 400 °C, respectively. Compressed air was used as heating gas (10 L/min) and argon was used as collision gas (270 kPa). Nitrogen was used as nebulizing gas (3 L/min) and drying gas (10 L/min). 

Multiple reaction monitoring (MRM) mode was applied for TTX and IS. TTX and its analogues were measured according to the transitions of [M+H]^+^ > [M+H-H2O]^+^ (23 eV) and [M+H]^+^ >162 (37 eV) [23]. The analogues included in this study were 11-oxoTTX ([M+H]^+^ 336), 4-epiTTX ([M+H]^+^ 320), 5-deoxyTTX or 11-deoxyTTX ([M+H]^+^ 304), 4,9-anhydroTTX ([M+H]^+^ 302), 11-norTTX-6(S)-ol or 11-norTTX-6(R)-ol ([M+H]^+^ 290), 5,6,11-trideoxyTTX ([M+H]^+^ 272), and 4,9-anhydro-5,6,11-trideoxyTTX ([M+H]^+^ 254) [18]. The MRM conditions are listed in Table 1.

### 2.5. Sample Preparation

The tissue was homogenized with a knife tissue homogenizer (Retsch GM 200, Retsch, Arzberg, Germany) for 2 min. About 2 g of the homogenized sample was weighed into a 15 mL polypropylene centrifugal tube. The sample was mixed on a vortex mixer (Muti Reax, Heidolph, Germany) with 8 mL of 1% HAc/50% methanol/water for 5 min and then conducted with ultrasound-assisted extraction for 20 min at room temperature. After that, the volume of the mixture was set to the 10 mL mark with 1% HAc/50% methanol/water and mixed for 0.1 min. The mixture was centrifuged at 8000 r/min for 5 min. An aliquot of 1 mL supernatant was mixed with 1.5 mL acetonitrile for 0.1 min and stayed at −18 °C for 10 min. Then, the mixture was centrifuged at 8000 r/min for 5 min and the supernatant was loaded onto a pretreated MCX SPE cartridge. The cartridge was washed with 3 mL of 0.2% HAc water and 3 mL of 0.2% HAc/50% acetonitrile/water successively after the upload finished. After that, the cartridge was vacuumed for 0.1 min and all the rinsed solvents were discarded. The analyte in the cartridge was then eluted with 3 mL of 0.2% HCl/50% acetonitrile/water. All the eluate was collected, spiked with 30 μL of IS spiking solution and 5 μL of ammonium, mixed for 0.1 min, and filtered by a Teflon membrane (0.2 μm) before LC-MS/MS measurement. The flow diagram for sample preparation is shown in Figure 2.

### 2.6. Preparation of Quality Control (QC) Samples

Based on the optimized analytical method, *N. didyma* and *N. semiplicatus* samples were measured and some of the positive samples were used as QC samples. The TTX concentration of the QC sample came from the average value of 6 parallel measurements. The concentrations of the prepared QC samples for the *N. didyma* matrix were 1.42 μg/kg, 207 μg/kg, and 2427 μg/kg, respectively. The values of the prepared QC samples for the *N. semiplicatus* matrix were 609 μg/kg, 1856 μg/kg, and 34,730 μg/kg, respectively. All the QC samples were stored at −18 °C.

### 2.7. Condition Optimization of Sample Extraction

QC samples of *N. didyma* (207 μg/kg) and *N. semiplicatus* (609 μg/kg) and TTX spiking *R. bezoar* samples (50 μg/kg) were used for the optimization of sample extract solvents. Each sample was conducted by ultrasound-assisted extraction with 1% HAc methanol, 1% HAc/50% methanol/water, and 1% HAc water at room temperature, respectively. The other steps were the same as mentioned in the Section 2.5 Sample preparation. 

Besides, another sample was processed by boiling-assisted extraction with 1% HAc water: About 2 g of the homogenized sample was weighed into a 15 mL polypropylene centrifuge tube. The sample was mixed with 8 mL of 1% HAc water for 5 min, and then incubated in boiling water for 5 min. After that, the sample mixture was cooled to room temperature with flowing tape water. The volume of the mixture was set to the 10 mL mark with 1% HAc water and mixed for 0.1 min. The next steps were the same as mentioned in the Section 2.5 Sample preparation.

### 2.8. Calculation of Matrix Effects

Matrix effects (ME) were calculated:$$\mathrm{ME}\%=\frac{100\times A_{\mathrm{matrix}}}{A_{\mathrm{solvent}}}$$
A_matrix_ is the peak area of TTX in the matrix and A_solvent_ is the peak area in pure solvent. The results more or less than 100% indicate matrix enhancement or suppression effects, respectively [24]. 

### 2.9. Graph making and Statistical Analysis

Statistical analysis was performed with SPSS 13.0 software (SPSS Inc. Chicago, IL, USA). Figures were generated using the Origin 8.0 (Origin Lab Inc. Northampton, MA, USA) program.

## 3. Results and Discussion

### 3.1. Sample Extraction

The extraction of TTX from seafood mainly included two methods: boiling-assisted extraction with HCl or HAc/water [15,16,17,22] and ultrasound-assisted extraction with an HAc/methanol/water mixture [19,20,23,25]. The boiling-assisted extraction method has a good effect on protein precipitation, but there is a loss in the recovery of TTX. Although the boiling time has decreased from 30 min [26] to 5 min [17,27], the method recoveries were only 70–84%. The recoveries of the ultrasound-assisted extraction method were 66.9–107% [19,20,23]. The recovery was related to the water content in the extract solvent mixture. The recovery results were a little to the low side with anhydrous extraction (dried product with HAc/methanol extraction) [23]. 

The recoveries of TTX in QC samples of *N. didyma* (207 μg/kg) and *N. semiplicatus* (609 μg/kg) and TTX spiking *R. bezoar* samples (50 μg/kg) with 4 extract solvents are shown in Figure 3. When extracted with 1% HAc methanol, the recoveries of TTX in three matrices were only 61.2–74.0%. When extracted with 1% HAc/50% methanol/water and 1% HAc water at room temperature, the results were 92.4–99.4%. The recoveries were reduced by about 10% under boiling-assisted extraction with 1% HAc water. When extracted with 1% HAc water at room temperature, the extract was easy to emulsify and produced foam, affecting constant volume operation. Therefore, ultrasound-assisted extraction with 1% HAc/50% methanol/water was used in this study.

### 3.2. Clean-Up by MCX Cartridge

#### 3.2.1. Protein Precipitation before MCX Cartridge Clean-Up

Acetonitrile was used as the protein precipitation reagent in this study. Under the optimal extraction conditions with 1% HAc/50% methanol/water for the QC samples of *N. didyma* (207 μg/kg) and *N. semiplicatus* (609 μg/kg) and TTX spiking *R. bezoar* samples (50 μg/kg), the effects of the volume of acetonitrile on the TTX recoveries are shown in Figure 4. The recoveries were only 74.1–79.5% if an aliquot of 1 mL of extract was directly loaded onto the MCX cartridge without using acetonitrile for protein precipitation (Condition No. 1, Figure 4). The results were less than 70% if an aliquot of 2 mL of extract was directly loaded (No. 2, Figure 4). 

Seafood, like gastropods, is rich in proteins in muscle and inorganic salts in viscus. Both will compete for the active sites of cation exchange in the MCX cartridge, affecting the recovery during SPE clean-up. The recoveries of TTX in each step during MCX cartridge clean-up (loading, washing with 0.2% HAc and 0.2% HAc/50% acetonitrile, eluting with 0.2% HCl/50% acetonitrile) are shown in Figure 5 for the QC sample of *N. didyma* (207 μg/kg) or the standard in the pure solvent mixture. TTX was not found during loading and washing steps if the standard in the pure solvent mixture was loaded. However, more than 20% or 30% of TTX was lost during loading and washing steps if an aliquot of 1 mL or 2 mL of sample extract was directly loaded (without precipitation of proteins and inorganic salts with acetonitrile), respectively. Therefore, it is necessary to select the appropriate extract solvent to minimize the contents of proteins and inorganic salts in the extract before MCX cartridge clean-up. 

Acetonitrile was used for further precipitation of protein and inorganic salts in the sample extract before MCX cartridge clean-up in this study. The recoveries of TTX in the studied three matrices were more than 90% if an aliquot of 1 mL of sample extract was added with 1 mL (No. 3 Figure 4) or 1.5 mL of acetonitrile (No. 4 Figure 4). TTX was not found during loading and washing steps (Figure 5) if an aliquot of 1 mL of sample extract was added with 1.5 mL of acetonitrile. The results slightly decreased if the volume of acetonitrile increased to 2 mL (No. 5 Figure 4). The solubility of TTX in water, methanol, and acetonitrile decreases sequentially. The increase in the acetonitrile content can improve the precipitation effects of protein and inorganic salts and improve the recovery during MCX cartridge clean-up. However, it will reduce the solubility of TTX in the extraction solution if the ratio of acetonitrile exceeds a certain value (No. 5, Figure 4). 

In summary, acetonitrile can further remove proteins and inorganic salts in the sample extract before MCX cartridge clean-up. The optimal ratio is that an aliquot of 1 mL of sample extract is added with 1.5 mL of acetonitrile. Satisfactory results of recovery for TTX will be obtained under this condition.

#### 3.2.2. TTX Eluting and the Treatment of Eluate

Normally, ammonia solution is used to elute the analyte from the cation exchange cartridge. However, TTX is unstable under alkaline conditions [6]. One to two milliliters of 0.1 or 0.2 mol/L HCl in methanol (about 1% or 2% HCl methanol) was used for the elution of TTX from the cartridge in literature [2,23]. This type of solvent mixture has strong elution strength, and severe matrix suppression effects were found for TTX in the eluate (less than 50%). In order to make it compatible with the subsequent separation of the LC column, it is also necessary to remove methanol and high concentrations of HCl by nitrogen drying. It is time-consuming and easy to corrode the nitrogen drying instrument as well.

TTX is a water-soluble substance. Its solubility in water is higher than that in methanol or acetonitrile. It is found in this study that increasing the proportion of water in the eluent can improve the elution ability of TTX. This makes it possible to reduce the concentration of HCl in the eluent. It was found that the optimal elution condition was 3 mL of 0.2% HCl/50% acetonitrile water. Furthermore, the pH value could be adjusted to greater than 3 when 5 μL of ammonia was added to the eluate from the MCX cartridge and meet the optimal pH requirement for the XBridge^TM^ BEH Amide column. Therefore, the eluate can be directly injected to LC-MS/MS without the nitrogen drying step.

### 3.3. Screening of Internal Standard

Chromatograms of TTX, IS, and voglibose are shown in Figure 6A, and matrix effects (*n* = 6) of TTX in *N. didyma*, *H. tuba*, *R. bezoar,* and *B. exarata* are illustrated in Figure 6B. The matrix effects of TTX in the studied 4 matrices were 86.9–92.2% analyzed by the optimized conditions. There was still about 10% signal suppression during LC-MS/MS measurement. Matrix matched standard calibration was required for the accurately quantitative determination of TTX in samples, which brings inconvenience to practical applications.

A stable isotope labeled internal standard calibration is the best solution to eliminate the influence of matrix effects in complex matrixes. However, there is no commercialized isotope labeled internal standard for TTX [28]. The key interfering factor of matrix effects for LC-MS/MS determination is the co-eluate during chromatographic separation. Therefore, when selecting non-isotopic internal standards, it is necessary to have a chromatographic retention time as close as possible between the internal standard and the analyte. Voglibose was used as an internal standard for the determination of TTX in blood. The retention time of voglibose differed from that of TTX by approximately 1 min [29,30]. Under the conditions used in this study, the difference in retention time between voglibose and TTX can be reduced to approximately 0.5 min. However, voglibose exhibited significant matrix enhancement effects in gastropod matrices (106–125%, Figure 6B). There was an obvious difference compared to the matrix suppression effects of TTX (86.9–92.2%). Voglibose is not suitable as an internal standard for quantitative correction of TTX in gastropods.

Screening the normally used drugs, it was found that validamycin has similar functional groups and physicochemical properties to TTX (Figure 1). Under the chromatographic separation conditions used in this study, the retention time of validamycin only differed by about 0.15 min from TTX (Figure 6A). The matrix effects of validamycin were almost consistent with those of TTX (88.1–94.6%, Figure 6B). The signal suppression effects of TTX during LC-MS/MS measurement can be effectively improved and the results can be calibrated with the standard in the solvent using the internal standard correlation with validamycin.

### 3.4. Selection of Chromatographic Column

TTX and IS belong to polar and hydrophilic substances containing multiple hydroxyl groups (Figure 1). The reversed C18 column has no retention capacity for TTX. An ion pair regent, such as sodium 1-heptanesulfonate or perfluorobutyric acid, is required for the LC-FLD determination of TTX [2,14]. Hydrophilic interaction chromatography (HILIC) columns can achieve a good separation effect of TTX without assistance from ion pair reagents. HILIC columns containing amide functional groups have the advantages of stable retention time, symmetrical chromatographic peaks, and high sensitivity. They have been widely used for the chromatographic separation of TTX [10,15,16,17,18,19,31]. The separation effects were compared between XBridge^TM^ BEH Amide columns (1.7 μm) with the specifications of 100 × 2.1 mm and 150 × 3.0 mm in this study. The retention time was about 2 min faster while the matrix effects were 70–84% if the former was used to separate TTX in gastropods. The matrix suppression effects separated with the former were much more severe than those separated with the latter (88.1–94.6%). Therefore, the XBridge^TM^ BEH Amide column (1.7 μm) with the specification of 150 × 3.0 mm was selected in this study.

### 3.5. Method Validation

#### 3.5.1. Limit of Detection (LOD) and Quantification (LOQ), and Linear Range

LOD was calculated according to the content corresponding to the signal-to-noise ratio (S/N) at 3 for the qualitative ion pair and LOQ based on S/N at 10 for the quantitative ion pair in the matrix. The LOD and LOQ were 0.5 μg/kg and 1 μg/kg, respectively.

Calibrated with the internal standard, the linear range was 0.1–100 μg/L (equivalent to 1.5–1500 μg/kg in the sample matrix). The equations of linear regression were y = 0.629677x + 0.007502, y = 0.602943x + 0.003966, and y = 0.600798x + 0.004648 with the correlation coefficient (*r*^2^) of 0.9998, 0.9991, and 0.9990 in pure solvent, *N. didyma,* and *R. bezoar*, respectively. The slope ratios in the equations of linear regression in *N. didyma* and *R. bezoar* relative to that in pure solvent were 0.96 and 0.95, respectively. Therefore, the concentration of TTX in the sample matrix can be quantitatively calibrated by the standard in pure solvent with the use of validamycin as the internal standard.

#### 3.5.2. Recovery

Three levels with six parallel measurements were done at 1.5, 44, and 1500 μg/kg in *N. didyma*, *H. tuba,* and *R. bezoar*. As shown in Table 2, the average recoveries in *N. didyma*, *H. tuba,* and *R. bezoar* were 83.0–92.5%, 82.6–94.4%, and 82.9–93.5% with the relative standard deviations (RSDs) of 4.7–8.4%, 3.8–8.0%, and 5.9–7.8%, respectively.

#### 3.5.3. Accuracy and Precision

QC samples of *N. didyma* (1.42 μg/kg, 207 μg/kg, and 2427 μg/kg, respectively) and *N. semiplicatus* (609 μg/kg, 1856 μg/kg, and 34,730 μg/kg, respectively) were used in this study. Six repetitions in the same day and three repetitions in three days, respectively, were done for measurements of intra-days’ and inter-days’ accuracy and precision. The eluate for QC samples with 2427 μg/kg of *N. didyma* or 1856 μg/kg of *N. semiplicatus* after MCX cartridge clean-up was 10-fold dilution with 50% ACN/water before measurement. The eluate for the QC sample with 34,730 μg/kg of *N. semiplicatus* after MCX cartridge clean-up was 50-fold dilution with 50% ACN/water before measurement. The results are presented in Table 3. The intra-day’s accuracy for TTX in *N. didyma* was 93.2–98.8% with the precision less than 7.4%. The inter-days’ accuracy for TTX in *N. didyma* was 92.4–97.8% with the precision less than 8.3%. The intra-days’ accuracy for TTX in *N. semiplicatus* was 91.8–97.4% with the precision less than 6.0%. The inter-days’ accuracy for TTX in *N. semiplicatus* was 92.0–93.6% with the precision less than 7.2%.

### 3.6. Comparing with the Reported LC-MS/MS Methods

LC-MS/MS has been applied for the determination of TTX and its analogues in fish, bivalve mollusks, gastropods, and cooked seafood as shown in Table 4. The method with LOD < 1 μg/kg was the clean-up technique of the immune affinity [19] or MCX (this study) SPE cartridge. However, TTX analogues cannot be measured with the clean-up technique of the former. According to the parameters of LOD, matrix effects, and recovery, good results were obtained with the use of the method developed in this study.

### 3.7. TTX in Gastropods

#### 3.7.1. TTX Contaminations in Gastropods

TTX contaminations in 741 samples with 7 species of gastropods sampled from 2016–2022 are presented in Table 5. Except for *N. semiplicatus*, the other six species of gastropods are commercially available seafood in *Zhejiang* province, China. TTX was not found in *H. tuba* and *R. clavigera* samples. The detection rates in *R. bezoar* and *B. exarata* samples were 3.2% and 6.7%, respectively, with concentrations less than 5.6 μg/kg. The detection rate in *M. pupillus* samples was 12.5% with concentrations less than 41.2 μg/kg. The concentrations of TTX in the above five species of gastropods were below the EFSA proposed provisional level of 44 μg TTX equivalents/kg in shellfish [9]. 

*N. didyma* (Figure 7A) is a gastropod and a delicious seafood in China. TTX contaminations were studied in 565 *N. didyma* samples from 2016 to 2022. The detection rate of TTX was 34.2% and the average concentration was 23.1 μg/kg (calculated as 1/2LOD if TTX was not detected). The concentrations of TTX in three samples were between 1000–2000 μg/kg, and in one sample exceeded 2000 μg/kg (2327 μg/kg). The concentrations of TTX in 8.1% of the samples exceeded the EFSA proposed provisional level of 44 μg TTX equivalents/kg in shellfish [9]. This level is based on the assessment of the intake of 400 g of shellfish meat. The intake for the meat of *N. didyma* is far below 400 g/day for the residents in China. The safety assessment for *N. didyma* needs to consider the actual intake. 

TTX was found in all the seven samples of *N. semiplicatus* gotten from the coast of *Zhejiang* province, China. The sample with a TTX content of 34,730 μg/kg came from a poisoning case. The clean-up with the immune affinity column can achieve accurate quantification of TTX for samples at low concentrations [19,20]. However, for samples with TTX contents exceeding the column capacity (usually 1000 ng), multiple dilutions and multiple clean-up treatments should be conducted to obtain accurate results. Using MCX cartridge clean-up, the results can be accurately quantified by a single clean-up and appropriate dilution of the eluate to let the TTX content within a linear range, even if the concentration reaches 34,730 μg/kg.

#### 3.7.2. Time Distribution of TTX Contaminations in *N. didyma*

TTX contamination in *N. didyma* sampled from February to November are shown in Figure 7B–D. Samples were not collected in January and December. TTX was not detected in samples collected in February and November. The samples with the highest detection rates (exceeding 50%) were found in June and July. The samples with the highest concentration levels were found from May to August. In summary, the samples with high contaminations of TTX occurred from May to August.

TTX was supposed to be produced by symbiotic bacteria (endogenous) or accumulated through the food chain (exogenous) [10,38]. The results proved that the origin of TTX was the latter, but not the former since the concentrations of TTX in samples of *N. didyma* changed significantly over time. The results also indicated the concentration change of TTX in the food of *N. didyma*. The high contamination of TTX in the food of *N. didyma* began in May and June. The concentration of TTX in *N. didyma* reached its highest level in June and July through the accumulation in the food chain. After August, the concentration of TTX in the food of *N. didyma* decreased, obviously. The metabolic rate of TTX in *N. didyma* was greater than the supplementation rate from the food chain. Therefore, the contamination of TTX decreased quickly. Except from May to July, the concentrations of TTX in *N. didyma* were relatively lower in the other months.

#### 3.7.3. TTX Analogues in *N. didyma* and *N. semiplicatus*

TTX analogues found in *N. didyma* and *N. semiplicatus* are shown in Figure 8. The analogues found in *N. semiplicatus* were 11-oxoTTX ([M+H]^+^ 336, 7), 4-epiTTX ([M+H]^+^ 320, 6), 5-deoxyTTX and 11-deoxyTTX ([M+H]^+^ 304, 5), 4,9-anhydroTTX ([M+H]^+^ 302, 4), 5,6,11-trideoxyTTX ([M+H]^+^ 272, 2), and 4,9-anhydro-5,6,11-trideoxyTTX ([M+H]^+^ 254, 1). The results were similar to those reported in the literature [39,40], except that a trace amount of 4,9-anhydro-5,6,11-trideoxyTTX ([M+H]^+^ 254, 1) was detected in this study. The analogues with 11-oxoTTX ([M+H]^+^ 336, 7), 4-epiTTX ([M+H]^+^ 320, 6), 5-deoxyTTX and 11-deoxyTTX ([M+H]^+^ 304, 5), 4,9-anhydroTTX ([M+H]^+^ 302, 4), 11-norTTX-6(S)-ol and 11-norTTX-6(R)-ol ([M+H]^+^ 290, 3), and 5,6,11-trideoxyTTX ([M+H]^+^ 272, 2) were detected in *N. didyma*. The analogues 11-norTTX-6(S)-ol and 11-norTTX-6®-ol ([M+H]^+^ 290, 3) were found in *N. didyma* but not in *N. semiplicatus*. On the contrary, 4,9-anhydro-5,6,11-trideoxyTTX ([M+H]^+^ 254, 1) was found in *N. semiplicatus* but not in *N. didyma*.

Relative potency (RP) was used for the toxicity evaluation of TTX and some of its analogues [9]. TTX has the highest toxicity among the analogues and its RP value was set at 1. The second most toxic analogue was 11-oxoTTX (7) and its RP value was 0.75. The RP values were less than 0.2 for the other analogues. As shown in Figure 8, the signal of 5-deoxyTTX/11-deoxyTTX (5) was about 10% compared with that of TTX in *N. didyma*. The signals of the other analogues were less than 1%. Considering the RP values, the toxicity of the analogues was less than 10% of TTX. The signal of 5,6,11-trideoxyTTX (2) was about 10% compared with that of TTX in *N. semiplicatus.* Both the signals of 5-deoxyTTX/11-deoxyTTX (5) and 11-oxoTTX (7) were about 5% of that of TTX. The total signals of the analogues were more than 20%. The toxicity of the analogues cannot be ignored in *N. semiplicatus*. It is necessary to conduct quantitative measurements of the analogues to evaluate their toxicity if the standard products are commercially available.

## 4. Conclusions

The established analytical method of TTX in gastropods by LC-MS/MS with MCX cartridge clean-up is simple and fast in operation. It is suitable for the determination of trace TTX in seafood and high concentrations in poisoning samples. The matrix suppression effects of TTX in samples can be compensated with the use of validamycin as the internal standard to ensure the accuracy of quantitative results. The contamination of TTX in *N. didyma* is firstly reported in this study. The results indicate that the high contamination of TTX in *N. didyma* was found from May to August. By studying the time distribution of TTX, it is inferred that the main source of TTX in *N. didyma* is from the accumulation of food chains (exogenous).

## Figures and Tables

**Figure 1 foods-12-03103-f001:**
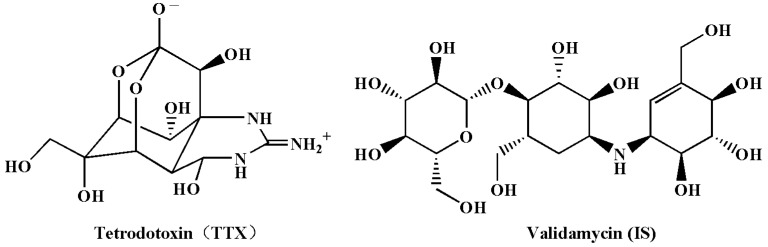
Structure of TTX and validamycin (IS).

**Figure 2 foods-12-03103-f002:**
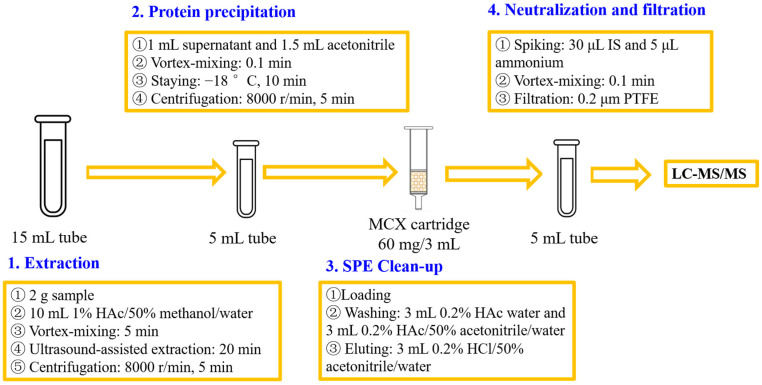
Flow diagram for sample preparation.

**Figure 3 foods-12-03103-f003:**
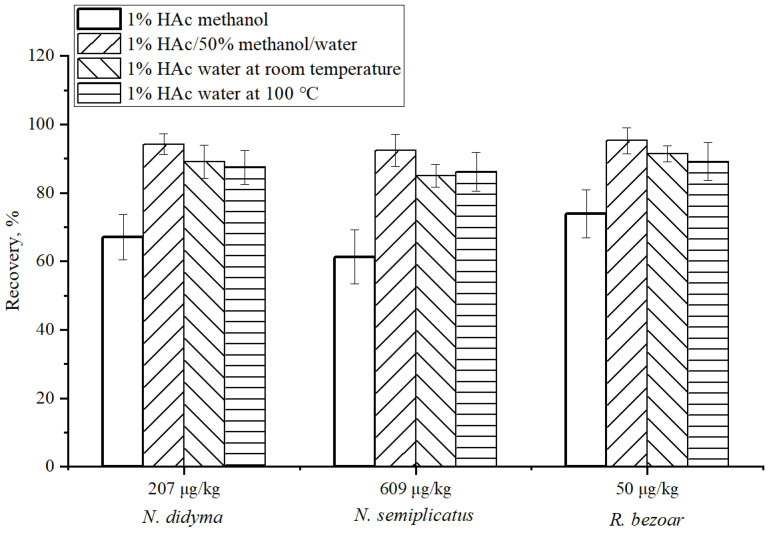
Recoveries of TTX in QC samples of *N. didyma* (207 μg/kg) and *N. semiplicatus* (609 μg/kg) and TTX spiking *R. bezoar* samples (50 μg/kg) extracted with 4 different extraction solvents.

**Figure 4 foods-12-03103-f004:**
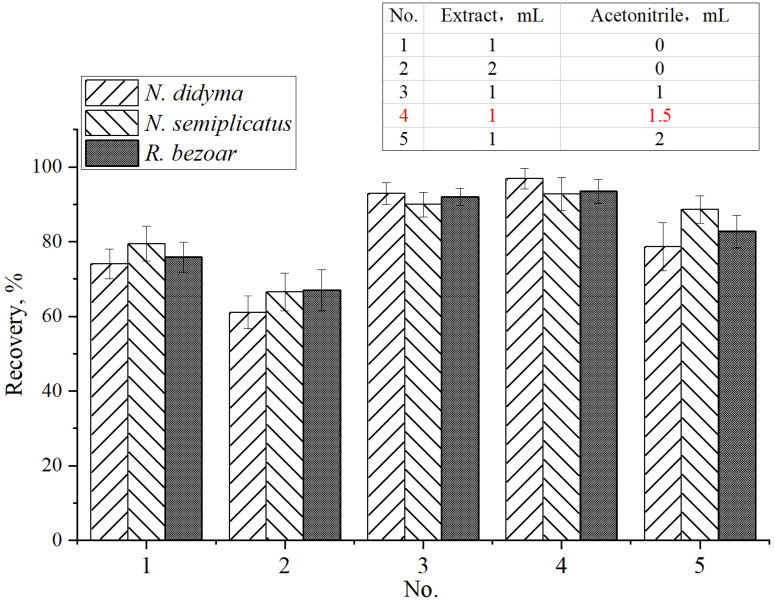
The effects of the volume of acetonitrile as the protein precipitation reagent on the TTX recoveries for the QC samples of *N. didyma* (207 μg/kg) and *N. semiplicatus* (609 μg/kg) and TTX spiking *R. bezoar* samples (50 μg/kg) extracted with 1% HAc/50% methanol/water.

**Figure 5 foods-12-03103-f005:**
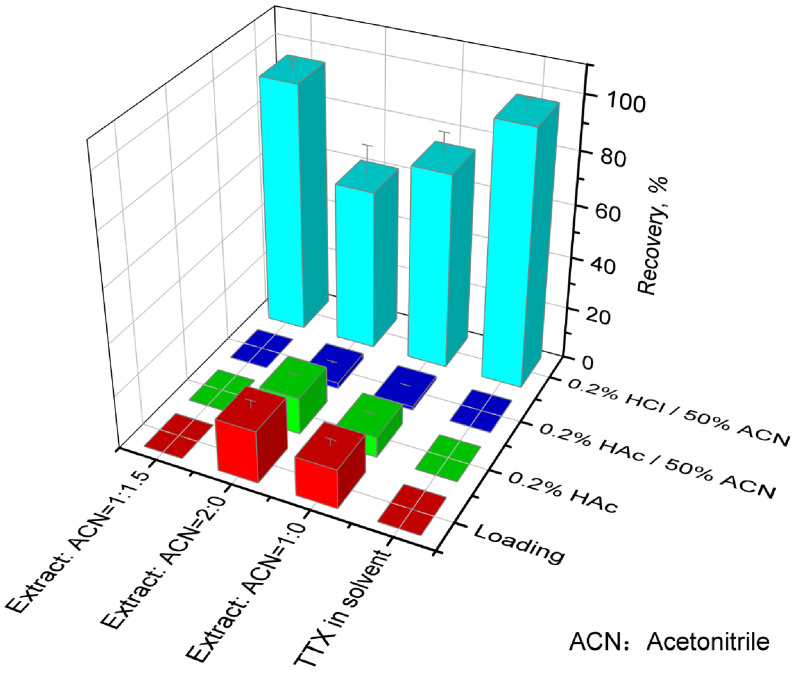
Recoveries of TTX in each step during MCX cartridge clean-up (loading, washing with 0.2% HAc and 0.2% HAc/50% acetonitrile, eluting with 0.2% HCl/50% acetonitrile) for QC samples of *N. didyma* (207 μg/kg) or the standard in the pure solvent mixture.

**Figure 6 foods-12-03103-f006:**
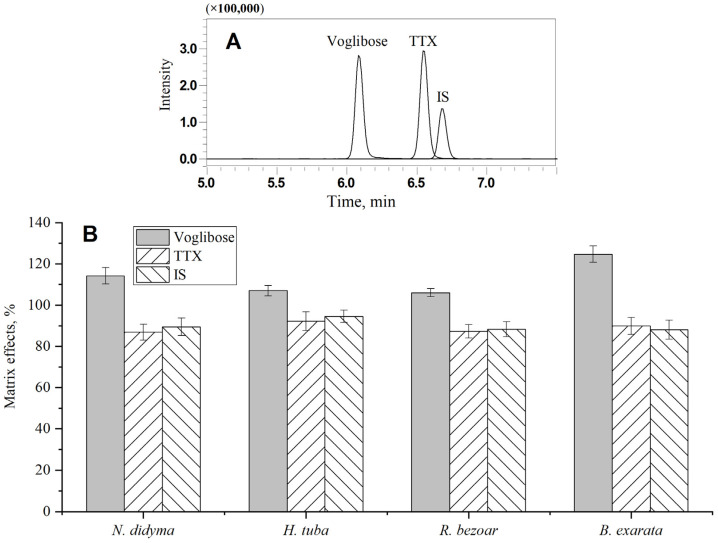
Chromatograms of TTX, IS, and voglibose (**A**), and matrix effects (*n* = 6) of TTX, IS, and voglibose in *N. didyma*, *H. tuba*, *R. bezoar,* and *B. exarata* (**B**).

**Figure 7 foods-12-03103-f007:**
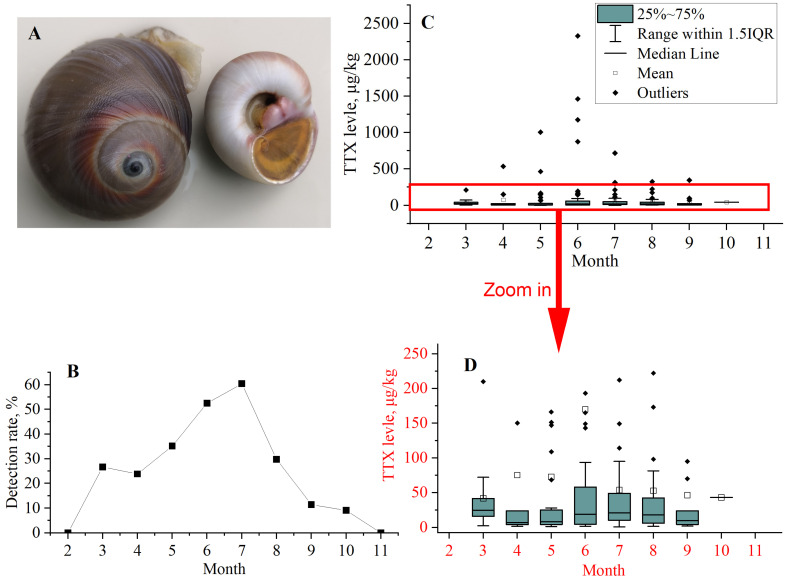
Time distribution of TTX contaminations in *N. didyma.* (**A**) The picture of *N. didyma*; (**B**) the detection rate; (**C**) the box chart of the TTX levels in different months; (**D**) zoom area of the TTX concentrations below 250 μg/kg in the box chart.

**Figure 8 foods-12-03103-f008:**
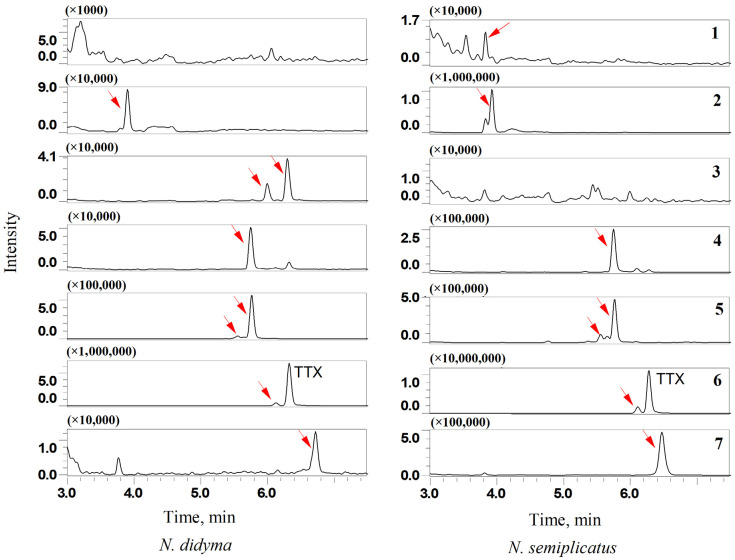
TTX analogues in *N. didyma* and *N. semiplicatus* with 4,9-anhydro-5,6,11-trideoxyTTX ([M+H]^+^ 254, 1), 5,6,11-trideoxyTTX ([M+H]^+^ 272, 2), 11-norTTX-6(S)-ol and 11-norTTX-6(R)-ol ([M+H]^+^ 290, 3), 4,9-anhydroTTX ([M+H]^+^ 302, 4), 5-deoxyTTX and 11-deoxyTTX ([M+H]^+^ 304, 5), 4-epiTTX ([M+H]^+^ 320, 6), and 11-oxoTTX ([M+H]^+^ 336, 7).

**Table 1 foods-12-03103-t001:** MRM conditions for TTX, TTX analogues, and IS.

TTXs	Precursor Ion (*m/z*)	Product Ion (*m/z*) ^a^	Collision Energy (eV)
TTX	320	302/162	23/37
IS	498	178/336	28/23
4,9-anhydro-5,6,11-trideoxyTTX	254	236/162	23/37
5,6,11-trideoxyTTX	272	254/162	23/37
11-norTTX-6(S)-ol and 11-norTTX-6(R)-ol	290	272/162	23/37
4,9-anhydroTTX	302	284/162	23/37
5-deoxyTTX and 11-deoxyTTX	304	286/162	23/37
4-epiTTX	320	302/162	23/37
11-oxoTTX	336	318/162	23/37

^a^ The first is quantification ion.

**Table 2 foods-12-03103-t002:** Recovery and RSD for TTX spiked in 3 sample matrices (*n* = 6).

Matrix	Spiking Level (μg/kg)	Recovery(%)						Average (%)	RSD (%)
		1	2	3	4	5	6		
*N. didyma*	1.5	81.2	87.4	81.6	80.6	93.8	73.4	83.0	8.4
	44	92.2	90.8	98.9	92.7	85.3	90.7	91.8	4.7
	1500	90.5	96.2	95.0	88.9	98.6	85.8	92.5	5.3
*H. tuba*	1.5	86.3	75.4	81.0	90.9	82.1	80.1	82.6	6.5
	44	92.3	95.4	89.8	97.8	92.1	88.3	92.6	3.8
	1500	95.0	104.3	88.7	92.8	101.3	84.2	94.4	8.0
*R. bezoar*	1.5	86.6	81.5	79.2	73.3	84.4	92.2	82.9	7.8
	44	88.6	97.3	94.4	85.6	87.0	98.0	91.8	5.9
	1500	92.0	95.8	105.8	86.4	92.0	88.8	93.5	7.3

**Table 3 foods-12-03103-t003:** Intra- and inter-days’ accuracy and precision for QC samples of *N. didyma* and *N. semiplicatus*.

Matrix	QC Levels (μg/kg)	Intra-Day (*n* = 6)		Inter-Days (3 days, *n* = 3)	
		Accuracy (%)	Precision (%)	Accuracy (%)	Precision (%)
*N. didyma*	1.42	93.2	7.4	92.8	8.3
	207	94.2	3.7	97.8	4.6
	2427 ^a^	98.8	4.0	92.4	5.8
*N* *. semiplicatus*	559	97.4	4.6	92.0	5.3
	1856 ^a^	95.5	5.5	92.2	5.3
	34,730 ^b^	91.8	6.0	93.6	7.2

^a^ 10-fold dilution with 50% ACN/water before measurement. ^b^ 50-fold dilution with 50% ACN/water before measurement.

**Table 4 foods-12-03103-t004:** Summary of TTX and its analogues’ pretreatment and analysis.

Toxins	Matrix	Extraction	Clean-Up	LC Column	LOD (μg/kg)	Matrix Effects (%)	Recovery (%)	Reference
TTX	Gastropod *Nassarius*	0.05 mol/L acetic acid, boiling, 30 min	Sep-Pak plus C18	AQ-C18	NM ^a^	NM	NM	[26]
TTX and 4 analogues	Puffer fish	0.05 mol/L acetic acid, boiling, 5 min	Sep-Pak C18, and activated charcoal	BEH Amide	160	NM	NM	[32]
TTX and 7 analogues	Puffer fish	0.05 mol/L acetic acid	Cleanert ODS	HILIC Silia	0.1 ng/mL	NM	94.2–108.3	[33]
TTX and 6 analogues	Cooked seafood	2% formic acid/methanol; ultrasound-assisted extraction, 20 min, RT	MCX, 0.1 M HCl in methanol	BEH Amide	5	80-110	66.9–89.2	[23]
TTX and 7 analogues	Puffer fish and trumpet shell	1% acetic acid, boiling, 8 min	Strata C18-E	ZIC HILIC	410	82.1	61.17	[34]
TTX and 3 analogues	Puffer fish	Water	Home-made SPME fibers	HILIC Silica	2.3	NM	NM	[35]
TTX and 4,9-anhydroTTX	Bivalve mollusks	0.1M HCl; boiling, 5 min	PCOX clean-up, dichloromethane extraction and hypercarb cartridge	Hypercarb	6.2–10.8	>89.4	90–119	[36]
TTX and 3 analogues	Bivalve mollusks	1% acetic acid, boiling, 5 min	ENVI-Carb	BEH Amide	7.84	45	68	[37]
TTX	Scallop and short-necked clam	1% acetic acid, boiling, 5 min	Ion-pair SPE	BEH Amide	27.4	95/97	75.7–78.9	[31]
TTX	Bivalve mollusks	1% acetic acid, boiling, 5 min	ENVI-Carb	BEH Amide	25	NM	NM	[16]
TTX	Fresh and heat-processed aquatic products	1% acetic acid/methanol; ultrasound-assisted extraction, 15 min, 40 °C	Immuneaffinity columns	BEH Amide	0.2	81.7	90.5–107.2	[19]
TTX and 8 analogues	Gastropod	1% acetic acid/50% methanol/water; ultrasound-assisted extraction, 20 min, RT	MCX, 0.2% HCl/50% acetonitrile/water	BEH Amide	0.5	86.9–92.2	82.6–94.4	This study

^a^ NM: not mentioned.

**Table 5 foods-12-03103-t005:** TTX contaminations in seven species of gastropods.

Gastropods	Sample Numbers	Positive Numbers	Detection Rate (%)	Concentrations (μg/kg)
*Hemifusus tuba*	30	0	0.0	ND ^a^
*Reishia clavigera*	30	0	0.0	ND
*Rapana bezoar*	62	2	3.2	ND~5.6
*Bullacta exarata*	30	2	6.7	ND~4.8
*Margarites pupillus*	16	2	12.5	ND~41.2
*Nassarius semiplicatus*	8	8	100.0	82.9~34,730
*Neverita didyma*	565	193	34.2	ND~2327

^a^ ND: not detected.

## Data Availability

The data presented in this study are available on request from the corresponding author.

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
