# Peer review of "Analytical Method Optimization of Tetrodotoxin and Its Contamination in Gastropods"

_foods, 2023, doi:10.3390/foods12163103_

Round 1

Reviewer 1 Report

In this study the authors optimized an analytical method for determination of tetrodotoxin and its contamination in gastropods. The subject is relevant and the manuscript fits within the scope of the journal.

The authors specified in section 2.5. that they used MCX SPE cartridge for sample preparation. Then in section 2.7 they presented two extraction methods: ultrasound-assisted extraction and boiling-assisted extraction for QC samples and in section 3 the authors said that they used MCX SPE cartridge for clean-up. It is not understood why they did not use SPE also for QC samples and why they chose the other two extraction methods, or they choose SPE only as clean up method. They compared recovery for ultrasound-assisted extraction and boiling-assisted extraction. If they used SPE for sample preparation, why they didn’t compare with the other two extraction methods used. The authors should clarify these aspects.

Please, check the whole manuscript carefully for typos for example in table 2 “accracy”.

Author Response

In this study the authors optimized an analytical method for determination of tetrodotoxin and its contamination in gastropods. The subject is relevant and the manuscript fits within the scope of the journal.

1. The authors specified in section 2.5. that they used MCX SPE cartridge for sample preparation. Then in section 2.7 they presented two extraction methods: ultrasound-assisted extraction and boiling-assisted extraction for QC samples and in section 3 the authors said that they used MCX SPE cartridge for clean-up. It is not understood why they did not use SPE also for QC samples and why they chose the other two extraction methods, or they choose SPE only as clean up method. They compared recovery for ultrasound-assisted extraction and boiling-assisted extraction. If they used SPE for sample preparation, why they didn’t compare with the other two extraction methods used. The authors should clarify these aspects.

Response: As shown in Figure 2 in this revised version, four steps were conducted in the optimal procedure of sample preparation in this manuscript. The former three steps are Step 1: ultrasound-assisted extraction with 1% HAc/50% methanol/water. Step 2: protein precipitation of the sample extract from Step 1. Step 3: SPE clean-up for the supernatant from Step 2. For optimization of extraction conditions, both of the extracts from ultrasound-assisted extraction and boiling-assisted extraction were treated with Step 2 and 3 (“The other steps were the same as mentioned in the section of 2.5 Sample preparation.” as presented in Section 2.7. ).

As mentioned in the 1st and 2nd paragraphs of Section 3.1., the method recoveries were only 70-84% (Reference 17 and 27) and the recoveries were reduced by about 10% (Figure 3 in this study) for boiling-assisted extraction. Therefore, there was no further SPE conditions optimization for the extract from boiling-assisted extraction.

As mentioned in the 1st paragraph of Section 3.2.1., QC samples were also used for the optimization of SPE conditions.

2. Please, check the whole manuscript carefully for typos for example in table 2 (Table 4 in this revised version) “accracy”.

Response: Thanks for your comments. “accracy” in Table 4 has been revised to accuracy.

The typos in the text was checked and revised.

Reviewer 2 Report

The research article "Analytical method optimization of tetrodotoxin and its contamination in gastropods" introduces the topic of the neurotoxin TTX in seafood and the necessity for potent analytical methods for its detection. The authors stepwise investigate the extraction, extract purification via SPE, and instrumental analysis to improve the analysis of TTX. The authors further mention the improved method covers derivatives of TTX.  

Dear authors,

Your article is methodologically sound, and all details are presented understandably. Please consider revision of the language; it will improve your work. Incorporating derivatives of TTX (3.6.3) was interesting. However, I´d advise adding information on the neurotoxic potential of the analogs. Are they similarly toxic or neglectable in the face of an overwhelming abundance of TTX?

Line 110. Dr, Ehrenstorfer, Augsburg, Germany 

Table 2. Accuracy

Please consider revising the language; it will improve your work and the reception of your article.

Author Response

The research article "Analytical method optimization of tetrodotoxin and its contamination in gastropods" introduces the topic of the neurotoxin TTX in seafood and the necessity for potent analytical methods for its detection. The authors stepwise investigate the extraction, extract purification via SPE, and instrumental analysis to improve the analysis of TTX. The authors further mention the improved method covers derivatives of TTX.  

Dear authors,

1. Your article is methodologically sound, and all details are presented understandably. Please consider revision of the language; it will improve your work. Incorporating derivatives of TTX (3.6.3, in this revised version is 3.7.3) was interesting. However, I´d advise adding information on the neurotoxic potential of the analogs. Are they similarly toxic or neglectable in the face of an overwhelming abundance of TTX? 

Response:

Thanks for your comments. The following discussions were added in this revised version in the last paragraph of Section 3.7.3.

Relative potency (RP) was used for toxicity evaluation of TTX and some of its analogues [9]. TTX has the highest toxicity among the analogues and its RP value was set with 1. The second most toxic analogue was 11-oxoTTX (7) and its RP value was 0.75. The RP values were less than 0.2 for the other analogues. As shown in Figure 8, the signal of 5-deoxyTTX/11-deoxyTTX (5) was about 10% comparing with that of TTX in N. didyma. The signals of the other analogues were less than 1%. Considering the RP values, the toxicity of the analogues was less than 10% of TTX. The signal of 5,6,11-trideoxyTTX (2) was about 10% comparing with that of TTX in N. semiplicatus. Both the signals of 5-deoxyTTX/11-deoxyTTX (5) and 11-oxoTTX (7) was about 5% of that of TTX. The total signals of the analogues were more than 20%. The toxicity of the analogues cannot be ignored in N. semiplicatus. It is necessary to conduct quantitative measurements of the analogues to evaluate their toxicity if the standard products are commercially available.

2. Line 110. Dr, Ehrenstorfer, Augsburg, Germany 

Response: revised.

3. Table 2. Accuracy

Response: revised.

4. Please consider revising the language; it will improve your work and the reception of your article.

Response: Thanks for your advice. The language is refined by the authors.

Reviewer 3 Report

The Author try to cover potential overview of tetrodotoxin with different analytical methods in manuscript. But there are following points need to correct in manuscript:

·         Please define one line of Tetrodotoxin in abstract to define clearly about your work.

·         In abstract focus more of the scientific achievements and make it shorter and effective.

·         The line no 14 “The effects of sample extraction…..  matrix was studied” is irrelevant to write in this section. Write it before or rewrite this sentence.

·         Write reference in Line no 29 “A global retrospective 29 study by Guardone et al.”

·         The line 31 is not clear “These kinds of 31 marine animals include” rewrite the sentence with specific name and types.

·         The line no 36” The contamination of TTX has become a hot topic in research” is not required in this sentence.

·         Citation of methodology is missing in materials and samples section check line 101

·         2.4. Instrumentation conditions: refer previous work has been done on same procedure and compare in this section.

·         Instead of sentence only use flow diagram also in methodology section for clarity and sequence of work.

·         The Toxic response of TTX contamination and effects can is not well mentioned in this manuscript.

·          The interpretation of Figure 7. TTX analogues is not well defined. Rewrite to justify the figure.

·         The discussion part of  manuscript is week. It is not corelate with previous study and analytical work.

·         Does author think about food safety and contamination of of TTX in N. didyma in conclusion section. Kindly Explain about food safety and other parameter to consider seafood in May to July.

Extensive English correction is required

Author Response

The Author try to cover potential overview of tetrodotoxin with different analytical methods in manuscript. But there are following points need to correct in manuscript:

  1. Please define one line of Tetrodotoxin in abstract to define clearly about your work.

Response:  Our work was define as “An analytical method was developed for both trace contamination and extremely high level of TTX in gastropods by liquid chromatography-tandem mass spectrometry (LC-MS/MS) with clean-up of cation exchange solid phase extraction (SPE) in this study.” in the Abstract.

  1. In abstract focus more of the scientific achievements and make it shorter and effective.

Response:  The 2nd and 3rd sentences were combined to one as “An analytical method was developed for both trace contamination and extremely high level of TTX in gastropods by liquid chromatography-tandem mass spectrometry (LC-MS/MS) with clean-up of cation exchange solid phase extraction (SPE) in this study.” The last sentence was revised to “The time distribution study indicated that high contaminations of TTX occurred from May to August for N. didyma.”

  1. The line no 14 “The effects of sample extraction…..  matrix was studied” is irrelevant to write in this section. Write it before or rewrite this sentence.

Response: This sentence was deleted.

  1. Write reference in Line no 29 “A global retrospective 29 study by Guardone et al.”

Response: The reference was add as “A global retrospective study by Guardone et al. [1] found that ……”

  1. The line 31 is not clear “These kinds of 31 marine animals include” rewrite the sentence with specific name and types.

Response: The sentence was revised as “The marine animals included puffer fishes, gastropods, arthropods and cephalopods.”

  1. The line no 36” The contamination of TTX has become a hot topic in research” is not required in this sentence.

Response: This sentence was deleted.

  1. Citation of methodology is missing in materials and samples section check line 101

Response: The analytical method was developed in this study.

  1. 2.4. Instrumentation conditions: refer previous work has been done on same procedure and compare in this section.

Response: The LC and MS conditions for TTX in this study were not reported. The MRM conditions for TTX analogues were referred from Reference [23] and the detailed components was from Reference [18].

  1. Instead of sentence only use flow diagram also in methodology section for clarity and sequence of work.

Response: The flow diagram for sample preparation is shown in a new Figure 2.

  1. The Toxic response of TTX contamination and effects can is not well mentioned in this manuscript.

Response: The toxic response of TTX was stated in the 2nd paragraph of Section 1. Introduction.

  1. The interpretation of Figure 7 (Figure 8 in this revised version). TTX analogues is not well defined. Rewrite to justify the figure.

Response: Thanks for your comments. A new paragraph was added in the end of Section 3.7.3. TTX analogues in N. didyma and N. semiplicatus.

  1. The discussion part of  manuscript is week. It is not corelate with previous study and analytical work.

Response: A new Section was added as 3.6. Comparing with the reported LC-MS/MS methods. Besides, detailed discussions were presented in the last paragraph of Section 3.7.3.

  1.  Does author think about food safety and contamination of of TTX in N. didyma in conclusion section. Kindly Explain about food safety and other parameter to consider seafood in May to July.

Response: The sentence “It is better to avoid these 3 months for the consume of this seafood.” in Conclusions was deleted. The food safety assessment for N. didyma was discussed in the 2nd paragraph in Section 3.7.1.

  1. Extensive English correction is required

Response: English was refined by the authors.

Reviewer 4 Report

See the attachment.

Author Response

In the manuscript entitled "Analytical method optimization of tetrodotoxin and its contamination in gastropods", authors validated an analytical method for the analyses of TTX in gastropods with an LC-MS/MS instrument. The manuscript is generally well written, and the topic suits the aims and scope of the Foods journal. However, in my opinion, there are typos and authors must discuss the analytical method better (and more) and compare it with other LC-MS/MS methods. Given these shortcomings, the manuscript requires major revisions.

  1. Originality: there are different analytical methods that analyze TTX in LC-MS/MS. Examples include:

1) Huang, H. N., Lin, J., & Lin, H. L. (2008). Identification and quantification of tetrodotoxin in the marine gastropod Nassarius by LC–MS. Toxicon, 51(5), 774-779.

2) Patria, F. P., Pekar, H., & Zuberovic-Muratovic, A. (2020). Multi-Toxin Quantitative Analysis of Paralytic Shellfish Toxins and Tetrodotoxins in Bivalve Mollusks with Ultra-Performance Hydrophilic Interaction LC-MS/MS—An In-House Validation Study. Toxins, 12(7), 452.

3) Chen, X. W., Liu, H. X., Jin, Y. B., Li, S. F., Bi, X., Chung, S., ... & Jiang, Y. Y. (2011). Separation, identification and quantification of tetrodotoxin and its analogs by LC–MS without calibration of individual analogs. Toxicon, 57(6), 938-943.

4) Jang, J. H., Lee, J. S., & Yotsu-Yamashita, M. (2010). LC/MS analysis of tetrodotoxin and its deoxy analogs in the marine puffer fish Fugu niphobles from the southern coast of Korea, and in the brackishwater puffer fishes Tetraodon nigroviridis and Tetraodon biocellatus from Southeast Asia. Marine Drugs, 8(4), 1049-1058.

5) Tsujimura, K., & Yamanouchi, K. (2015). A rapid method for tetrodotoxin (TTX) determination by LC-MS/MS from small volumes of human serum, and confirmation of pufferfish poisoning by TTX monitoring. Food Additives & Contaminants: Part A, 32(6), 977-983.

6) Bane, V., Hutchinson, S., Sheehan, A., Brosnan, B., Barnes, P., Lehane, M., & Furey, A. (2016). LC-MS/MS method for the determination of tetrodotoxin (TTX) on a triple quadruple mass spectrometer. Food Additives & Contaminants: Part A, 33(11), 1728-1740.

7) Patria, F. P., Pekar, H., & Zuberovic-Muratovic, A. (2020). Multi-Toxin Quantitative Analysis of Paralytic Shellfish Toxins and Tetrodotoxins in Bivalve Mollusks with Ultra-Performance Hydrophilic Interaction LC-MS/MS—An In-House Validation Study. Toxins, 12(7), 452.

Why your analytical method is better? Compare your analytical method with other methods. Especially for biotoxins, it is essential to have a multi-residue analysis. Why it is essential your method? Because if it is similar to others, no need a new manuscript even if the scientific quality is high.

Response: Thanks for comments. A new Section was added as 3.6. Comparing with the reported LC-MS/MS methods.

  1. L16: Why is “validamycin” in caps?

Response: Validamycin was revised to validamycin.

  1. L41: What is “MU”?

Response: MU/g was revised to mouse units/g (MU/g)

  1. L50-L53: “Mouse bioassay…confirmation”.

Response: The sentences were revised to “MBA is a classic method for TTX at mg/kg level in poisoning sample while the sensitivity is insufficient for TTX at μg/kg level in seafood. Immunoassay can quickly screen TTX in seafood, but its shortcoming in qualitative confirmation cannot be ignored as well”

  1. L64-L66: “However we found…detector”. This is not a scientific statements. Support the sentence with references or this is speculative.

Response: This sentence was deleted in the revised version since there are no references stated this issue. However, we still retain our views that this method is not suitable for routine application since three FLDs from different companies (Agilent, Waters and Shimadzu) has been damaged in our lab. The technical engineers also confirmed that their FLD cannot resistant to 4N NaOH. We proposed this statement is to inform readers to use this method with caution. We would not suffer the loss if there had been literature mentioning this issue before.

  1. L102-L106: Name of species must be in italic.

Response: Revised. We also have checked and revised the same problem in the whole text.

  1. L137-L145: I suggest a Table for the transition.

Response: Thanks for your suggestion. A new Table 1 was added.

  1. L149: How was vortexed? With which machine?

Response: All the word “vortex-mixed” was revised to “mixed”. The machine was added as “mixed on a vortex mixer (Muti Reax, Heidolph, Germany)”

  1. L150: How was centrifuged?

Response: The use of “centrifuged” please see the Section of 2.3. in Reference 19 in this manuscript.

  1. L161: Specify acronym (PTFE)

Response: PTFE was revised to Teflon.

  1. L186: Add equations how you calculate matrix effects. There is the template for equations for MDPI’s papers.

Response: The equation for matrix effects calculation was added as .

  1. L135: Specify the acronym “CID”

Response:  CID gas was revised to collision gas.

  1. L260: “However…conditions”. Add references.

Response: Reference [6] was added.

  1. L375-L377: “N….value” do you have any references for this? It has a scientific value?

Response: N. didyma is a delicious seafood in China. The sentences were deleted since there are no references found in English.

  1. Table 3 (Table 5 in this revised version): Add what ND means.

Response: A note was added as “a ND: not detected.” at the end of Table 5.

  1. Missing witch software do you use to do graphs and statistical analyses.

Response: A new Section of “2.9. Graph making and statistical analysis” was added.

  1. Compare the analytical parameters of your method with others analytical methods.

Response: A new Section was added as 3.6. Comparing with the reported LC-MS/MS methods.

Round 2

Reviewer 3 Report

Changes are done significantly, some minor english editings are still required.

minor editings required, grammar and punctuation

Reviewer 4 Report

The authors answered all the comments, therefore the manuscript can be accepted in its present form.